# Genome-wide association analysis of anthracnose resistance in sorghum [*Sorghum bicolor* (L.) Moench]

**Girma Mengistu**[1,2]*, **Hussein Shimelis**[1], **Ermias Assefa**[3], **Dagnachew Lule**[2]

**1** School of Agricultural, Earth and Environmental Sciences, University of KwaZulu-Natal, Scottsville, Pietermaritzburg, South Africa, **2** Oromia Agricultural Research Institute, Addis Ababa, Ethiopia, **3** Ethiopian Biotechnology Institute, Bioinformatics and Genomics Research Directorate (BGRD), Addis Ababa, Ethiopia

* germame2004@gmail.com

**Data Availability Statement:** All relevant data are within the paper and its Supporting information files.

## Abstract

In warm-humid ago-ecologies of the world, sorghum [*Sorghum bicolor* (L.) Moench] production is severely affected by anthracnose disease caused by *Colletotrichum sublineolum* Henn. New sources of anthracnose resistance should be identified to introgress novel genes into susceptible varieties in resistance breeding programs. The objective of this study was to determine genome-wide association of Diversity Arrays Technology Sequencing (DArTseq) based single nucleotide polymorphisms (SNP) markers and anthracnose resistance genes in diverse sorghum populations for resistance breeding. Three hundred sixty-six sorghum populations were assessed for anthracnose resistance in three seasons in western Ethiopia using artificial inoculation. Data on anthracnose severity and the relative area under the disease progress curve were computed. Furthermore, the test populations were genotyped using SNP markers with DArTseq protocol. Population structure analysis and genome-wide association mapping were undertaken based on 11,643 SNPs with <10% missing data. The evaluated population was grouped into eight distinct genetic clusters. A total of eight significant (P < 0.001) marker-trait associations (MTAs) were detected, explaining 4.86–15.9% of the phenotypic variation for anthracnose resistance. Out of which the four markers were above the cutoff point. The significant MTAs in the assessed sorghum population are useful for marker-assisted selection (MAS) in anthracnose resistance breeding programs and for gene and quantitative trait loci (QTL) mapping.

## Introduction

Sorghum [*Sorghum bicolor* (L.) Moench, 2n = 2x = 20] is an important cereal crop cultivated globally for multiple uses [1]. It is a mainstay crop in arid and semi-arid agro-ecologies due to its relatively higher drought tolerance compared to other common cereal crops such as maize and wheat. Various constraints affect sorghum production and productivity, notably by biotic stresses such as diseases, weeds (*Striga* species), and insect pests. Anthracnose, grain mold, leaf

**Funding:** National Research Foundation (NRF)/ South Africa for financial support for the research activity and the first author is postdoctoral in University of KwaZulu-Natal (UKZN).

**Competing interests:** The authors declare that they have no competing interests.

blight, rust and smut are the most important diseases of sorghum, while stem borer, shoot fly, termites and birds are the common insect pests attacking the crop [2–4].

Sorghum anthracnose caused by the fungal pathogen *Colletotrichum sublineolum* Henn. (previously known as *C. graminicola* [Ces.] G.W. Wilson) is amongst the most important diseases of the crop. Anthracnose causes yield and quality losses in sorghum production. For instance, 50–70% of grain yield loss has been reported in susceptible sorghum varieties in Ethiopia's drier and humid agro-ecologies [5,6]. The development and deployment of anthracnose resistant sorghum varieties is an economical and environmentally friendly approach that could bolster sustainable production and productivity. Understanding the genetic basis and dissection of genes conditioning anthracnose resistance using genomic tools enhances gene introgression and marker-assisted selection [4,7–10].

Genome-wide association study (GWAS) has been widely used to resolve complex genes controlling economic traits in crop plants, including anthracnose resistance for effective breeding and genetic analysis [4,7,8,11]. The genome-wide genetic analysis enables to the delineation of genomic regions such as markers, genes or quantitative trait loci (QTL) associated with crucial component traits for marker-assisted breeding, gene discovery or gene introgression [7,9]. Genome-wide association analysis depends on marker-trait association (MTA) using representative markers and genetically diverse test populations, including landraces, advanced breeding lines, and improved cultivars [12,13]. Identification of marker to trait associations will enhance gene introgression and selection gains through marker-assisted breeding.

Single nucleotide polymorphisms (SNP) are a marker of choice for genetic diversity analysis, characterization of genetic resources, cultivar identification, heterotic grouping, construction of high-resolution genetic maps, linkage disequilibrium-based association mapping and genetic diagnostics [14]. SNPs have been used in GWAS of various traits in sorghum, including genetic analysis for resistance to diseases such as anthracnose [4,7–9], grain mold [2], downy mildew, and head smut [4] and tolerance to drought [15]. Diversity Arrays Technology Sequencing (DArTseq) based single nucleotide polymorphisms (SNPs) is a powerful genomic tool used in GWAS of diverse agriculturally important crop traits [16].

Ethiopia is the centre of origin and diversity of both cultivated and wild sorghum [17–20]. The country maintains more than 9000 sorghum accessions through the Ethiopian Biodiversity Institute (EBI) [12]. Ethiopia's sorghum genetic resources have been used globally in genetic analysis and cultivar development programs [2,12]. Some novel genes conditioning resistance to grain mold [2] and anthracnose [8] were previously identified in the Ethiopian sorghum. The sorghum genetic resources maintained by the EBI should be systematically explored and used in crop improvement programs.

The sorghum genome was previously sequenced as a foundational database to guide genetic analysis and breeding programs [21]. Furthermore, various studies reported sorghum genomic regions associated with genes governing the inheritance of different traits such as anthracnose resistance [4,7,9,10,22], male sterility [12], flowering and plant height [12], heat tolerance [15] and grain polyphenol content [13]. Quantitative trait loci (QTL) conditioning protein, fat, and starch contents were reported in sorghum [23]. There are limited genomic studies examining marker-trait associations, including anthracnose resistance with a diverse and representative genetic pool of sorghum in Ethiopia.

The Bako areas in western parts of Ethiopia are known to maize and sorghum production. The same areas were reported as hotspot sites for sorghum anthracnose owing to their warm temperatures and high relative humidity [5]. To initiate an anthracnose resistance breeding program, many sorghum genetic resources were collected by the Ethiopian Biodiversity Institute (EBI) from nine regions in Ethiopia. The Melkassa Agricultural Research Center/Ethiopia preserved these collections. The genetic resources should be profiled using a high-density SNP

marker and anthracnose resistance to identify and introgress novel genes into susceptible varieties by resistance breeding programs to enhance the productivity of the crop. In light of the above background, this study's objective was to determine the genome-wide association of Diversity Arrays Technology Sequencing based single nucleotide polymorphism markers and anthracnose resistance genes in diverse sorghum populations of Ethiopia for resistance breeding.

## Materials and methods

### Plant materials

The study used 366 sorghum collections sourced from different geographical locations in Ethiopia, including the Afar, Amhara, Benshangul Gumuz, Dire Dawa, Gambella, Oromiya, SNNP, Somali, and Tigray regions acquired from Melkassa Agricultural Research Center in Ethiopia. Three improved varieties ('Gemedi', 'Geremew' and 'Btx623') were included as comparative controls. Btx623 is anthracnose susceptible variety obtained from Texas A and M University, USA. The descriptions of the test accessions are presented in Table 1.

### Study site and experimental design

The genotypes were evaluated for anthracnose resistance at Bako (9º6' N; 37º9' E) Agricultural Research Center (BARC) in Ethiopia. The centre receives an annual rainfall of 1,600 mm, while the mean maximum and minimum temperatures are 29 ºC and 13 ºC, respectively. The mean monthly relative humidity varies from 46 to 57%, while the main rainy season ranges from May to October, with the most rain received in July and August. Short rains are also received from March to June. The trends of weather data for BARC during the study periods (2016–2018) are summarized in Fig 1. The soil type of the study site is nitosol. The genotypes were established using a 61 x 6 row by column incomplete block design with three replications. Each plot consisted of a single row of 2.1 m long with inter-row and intra-row spacing of 75 cm and 15 cm, respectively.

### Pathotyping for anthracnose reaction, data collection and analysis

Based on preliminary evaluations, a single virulent anthracnose spore sourced from the Bako area was aseptically isolated, multiplied, and used to inoculate plants on 45 days after sowing using the procedure of Prom et al. (2019) [24]. The 366 genotypes were rigorously screened for anthracnose resistance at Bako for three seasons (2016–2018). Data on percentage severity of leaf area damaged by the anthracnose were recorded beginning from 15 days after inoculation five times at 10 days' intervals from five randomly tagged plants. The percentage of total leaf area of plants damaged by anthracnose were recorded following [25]. The final disease rating was measured 55 days after inoculation and was referred to as a final anthracnose severity (FAS). FAS was used to distinguish the anthracnose reaction of lines based on the level of severity. Mean severity percentage values for each plot were used for data analysis.

Disease severity for anthracnose was used to calculate the area under the disease progress curve (AUDPC). The AUDPC were calculated for each sorghum accession based on [26] as follows:

$$\text{AUDPC} = \sum_{i=1}^{n-1} \left[ (X_i + X_{i+1})/2 \right] (t_{i+1} - t_i)$$

The AUDPC values were converted into the relative area under the disease progress curve (rAUDPC) as a ratio of the actual AUDPC of a sorghum entry to the AUDPC of a susceptible

**Table 1. List of 366 sorghum genotypes used in this study.**

| Collection regions or research centres | Accessions |
|---|---|
| | Landraces |
| Afar | 72564, 72998, **73003**, 73006, 73007, **73008**, **73019**, **73026**, 73643, 73645, 206210, 206212 |
| Amhara | 69252, **70376**, 72443, 72467, **72474**, 72477, **72520**, **72524**, 72526, 72616, 73037, 73041, 73042, 73045, 73048, 73049, 73074, **73079**, **73095**, 75274, 75455, 200539, 210945, 210949, 210971, 210974, **211237**, **212640**, 213354, 214845, 214852, 226047, 226048, 226054, 226057, 228112, **228115**, 229887, 239154, 239179, 239180, 239182, **239184**, 239187, 239188, 239194, 239197, 239219, 239228, 239250, 242052, 243645, 243650, 243657 |
| Benshangul | SC283-14, **ETSL100375**, **PML981475**, 229832 |
| Dire Dawa | 70742, 71161, 71180, 228840, 239114, **239115**, 239116, 239117, 239118, 239119, 239123, 239124, 239125, 239126, **239127**, 239129, **239131**, 239132, 239133, 239134, **239135**, 239137 |
| Gambella | 69372, 69412, 70027, 70028, 70051, 71569, 71570, 71571, 71574, 71623, 71624, 71625, 71628, 71631, 71635, 71642, 71643, 71644, 71648, 71653, 71654, 71656, 71658, 71698, **71700**, 71701, 71708, 71711, 71712, 71714, 71720, 74914, 200522, 201433, 206149, 206154, 211209, 211210, **222885** |
| Oromiya | 9110, **9116**, 15830, **15832**, **15877**, 15890, 15897, 15904, **15908**, 15914, 15932, 15935, 15956, **16113**, 16116, 16133, **16135**, 16152, 16162, **16163**, 16168, 16171, 16173, 16176, 16177, 16180, 16206, 16208, 16212, 16213, 16440, 16450, 16451, 16477, 16487, 16489, 17518, 69534, **69540**, 69553, 70282, 70471, 70704, 70842, 70859, 70943, 70967, 70998, 71044, 71110, 71137, 71154, 71165, 71168, 71169, 71177, 71194, 71319, 71334, **71337**, 71363, 71372, 71374, 71392, **71395**, 71466, 71500, 71502, 71503, **71507**, 71513, 71516, 71524, **71544**, 71545, 71546, 71547, 71548, 71549, **71550**, 71551, 71553, 71555, 71556, 71557, 71558, 71559, 71560, 71562, **71563**, 75003, 75004, **75006**, 75114, 75115, **75118**, 75119, 75120, 75123, 75132, 75143, 75146, **75147**, 200126, 200193, 200306, 200307, 200308, 208740, 211251, 213201, **214110**, 223552, 223562, 228179, 228916, 228920, 228922, **234858**, 237550, 237804, 241221, 241265, 241267, 241282, 241283, 245062 |
| SNNP | 69088, 69178, 69326, 70161, 70187, 70636, 70795, 70874, 70891, 74649, 74651, **74653**, 74656, 74663, 74665, 74666, 74670, 74681, 74685, 74686, 74687, 204622, 204626, 204631, 204633, 204636, 210903, 210906, 241706, 241708, 241709, 241715, 241720, 241721, 241722, **241723**, **241725**, 241728 |
| Somali | 70436, 70844, **70864**, 231179, 231199, 231201, **231204**, 231458 |
| Tigray | 19613, 19619, 19621, 19641, 31309, 71420, 71424, 71425, 71476, 71479, 71480, 71484, 71489, **71497**, 73799, 73802, 73805, 73955, 73963, 73964, 74061, **74101**, 74130, 74133, 74145, **74157**, 74168, 74177, 74181, 74183, 74191, 74203, 74220, 74222, 74225, 74231, **74933**, 207876, 220014, **234088**, 234112, 235468, 237300, 238388, 238391, 238392, 238394, 238396, 238397, 238403, 238405, 238408, 238425, 238428, **238445**, 238449, 238450, 242043, 243670 |
| | **Improved varieties** |
| Bako ARC | Gemedi |
| Melkassa ARC | Geremew |
| Texas A&M University/USA | **BTx623** |

SNNP = Southern Nations, Nationalities, and Peoples', ARC = Agricultural Research Center.

* Bold face entries were not genotyped.

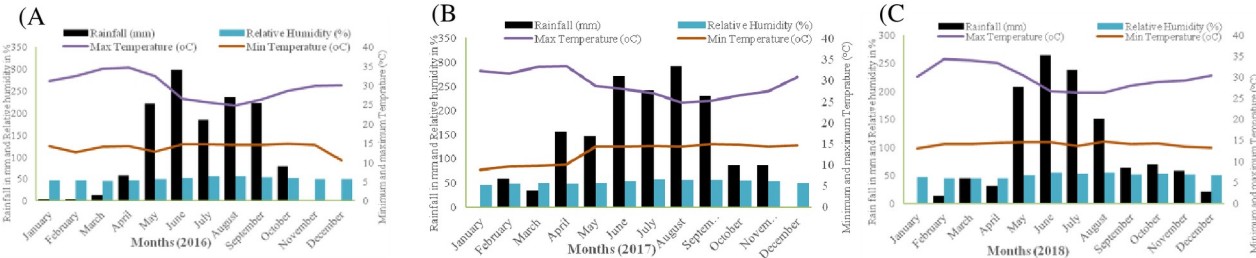

**Fig 1. Rainfall, relative humidity, and minimum and maximum temperatures of Bako Agricultural Research Center in 2016 (A), 2017 (B) and 2018 (C).** (Source: Bako Agricultural Research Center/Ethiopia).

landrace (Acc#239182) across the three cropping seasons (months May-October). The data were checked for normality, homogeneity of variance and validity for analysis of variance following the Bartlet test [27]. Data collected were analyzed using SAS computer software [28].

## DNA extraction and DArT sequencing

Due to the poor germination rate, 313 sorghum genotypes were assayed for genomic analysis out of the 366 genotypes. Test genotypes were planted in a greenhouse at Holeta National Agricultural Biotechnology Research Center using seedling trays. Eight leaf disc samples were collected from fresh and young leaf tissue using a 4×96 deep well sample collection plate three weeks after seedling emergence and sent to the Biosciences eastern and central Africa (BecA) hub of the International Livestock Research Institute (BecA-ILRI) in Kenya. Genomic DNA was extracted at BecA-ILRI/Kenya following the plant DNA extraction protocol for DArT [29]. The quality of DNA was checked for nucleic acid concentration and purity using a Nano-Drop 2000 spectrophotometer (ND-2000 V3.5, NanoDrop Technologies, Inc). Samples were genotyped using the DArTseq protocol with 24,634 SNP markers. After eliminating the DArT loci with unknown chromosome positions and filtering markers with more than 10% missing data, a total of 11,643 markers distributed across the 10 chromosomes were maintained for analysis.

The markers used had SNPs with call rate > 97%, and allele-calling equal or greater than 98% were selected. Genotypes with read depth less than the threshold were coded as missing. SNP markers have high rates of genotype missingness (>10%) and rare SNPs with <5% minor allele frequency (MAF) were discarded.

## Marker-trait data analysis

**Population structure.** Principal component analysis (PCA) was computed in the R package ggplot2 [30]. Model-based maximum likelihood analysis of population structure was calculated using the ADMIXTURE program, a high-performance tool for estimating ancestry in unrelated individuals [31]. To discern the optimum number of ancestral populations, ADMIXTURE was run with a 10-fold cross-validation procedure for K values varying from 2 to 20. The K value with the lowest standard error was selected. The graphical representation of the admixture patterns was depicted using the R package pophelper [32]. A membership coefficient greater than 0.70 discerned if a genotype is belonged to the group, as stated by Ketema et al. (2020) [33]. Admixed genotypes had a membership coefficient of less than 0.70 at each assigned K.

## Marker–trait association

Data on anthracnose severity and GBS based on 11,643 robust SNP markers were used for GWAS analysis. The FarmCPU Model [34] was used to perform an association analysis using the R software's GAPIT package [16]. The model splits the mixed linear model into a fixed effect and a random effect model to minimize false negatives caused by confounding population structure and SNPs. A coancestry matrix from ADMIXTURE was included as a covariate in GAPIT to reduce spurious associations.

The GWAS results were visually examined using the Manhattan and quantile-quantile (QQ) plots. The plots were generated using the CMPlot R package [35]. The cutoff of significant association was a False Discovery Rate (FDR) adjusted p-value less than 0.1, which was computed using the Benjamini and Hochberg procedure to control for multiple testing [36] and an exploratory significance cutoff p <0.001 was also used. SNPs with a high probability of

contribution to essential traits were tracked to the specific chromosome location based on the sorghum reference genome sequence, version 3.1.1 available at the Phytozome v12 [37].

## Identification of candidate genes

Candidate genes were identified based on the significant SNP markers associated with the traits. This was achieved using the physical genome assembly of sorghum reference genome sequence, version 3.1.1 (https://phytozome.jgi.doe.gov/pz/portal.html#!info?alias=Org *Sbicolor*) serving as identifying candidate genes between 20 kb on either side of significant SNPs. The putative functional candidate genes that co-localized with associated SNPs were annotated based on similarity to known annotated genes in other species such as Arabidopsis and tomato.

## Results

### Anthracnose resistance

Combined analysis of variance indicated significant differences (P < 0.01) due to the effects of genotype, season and genotype by season interaction for FAS and rAUDPC. These suggested the presence of considerable genetic variability among the test populations to select anthracnose resistant lines and aiding marker-trait analysis (Table 2).

There were 32 genotypes that scored lower rAUDPC values at the final stage of severity rating that ranged between 29.4 (entry 74685) to 40.9 (223562). These lines were identified as moderately resistant, making them ideal selections for anthracnose resistance breeding. Conversely, 334 genotypes were susceptible and expressed higher rAUDPC values ranging between 43.6 and 91.0. The rAUDPC showing the progression of disease severity on selected resistant and susceptible test genotypes during severity assessment intervals is presented in Fig 2. Four genotypes (239182, 72564, 73048 and 238428) scored the highest rAUDPC values compared with the anthracnose susceptible check (Btx623). The mean values for final severity and rAUDPC are presented in S1 Table.

### Marker characterization

**Population structure.**   Population structure analysis resolved eight sub-populations based on SNPs profiles. The Admixture algorithm was used to determine the population structure of the 313 sorghum genotypes. The ADMIXTURE analysis was performed for different K values varying from 2 to 20. Cross-validation error estimates for the ADMIXTURE models steeply

**Table 2.  Mean squares and significant tests for final anthracnose severity (FAS) and relative area under disease progress curve (rAUDPC) amongst 366 sorghum lines assessed in three seasons (2016–2018) at Bako Agricultural Research Center in Ethiopia.**

| Source of variation | DF | Parameters | |
|---|---|---|---|
| | | FAS | Raudpc |
| Seasons | 2 | 1728** | 27859** |
| Replication in season | 2 | 59ns | 5824** |
| Genotype | 365 | 1402** | 1620** |
| Genotype x season | 730 | 251** | 290** |
| Error | 2194 | 51 | 71 |

DF = degree of freedom, FAS = Final anthracnose severity, rAUDPC = Relative area under disease progress curve, ** denote significant difference at P < 0.01and ns = non-significant.

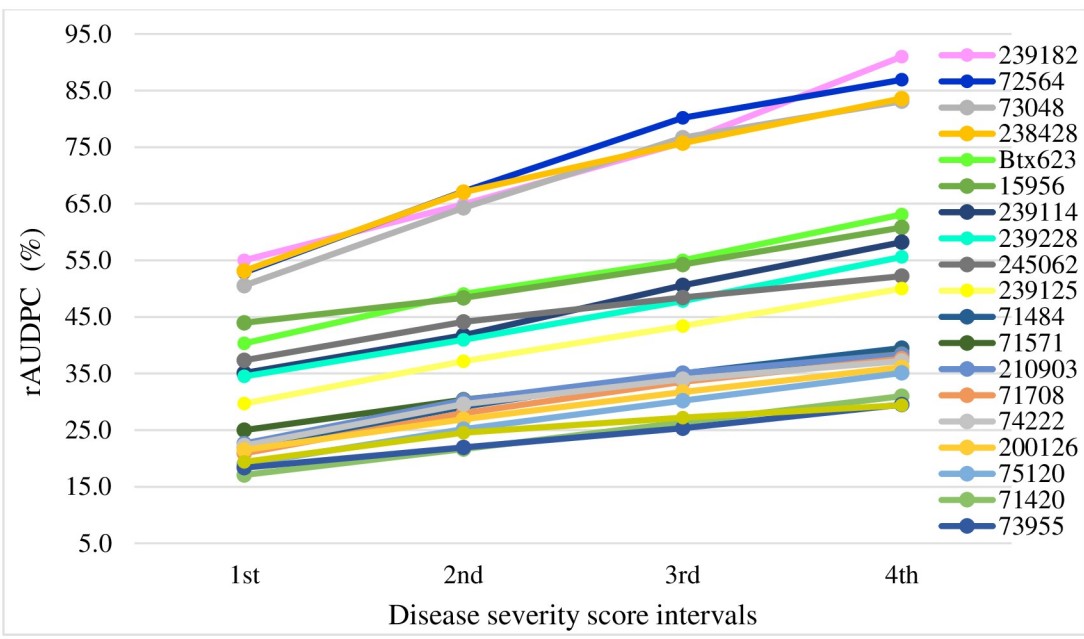

**Fig 2. The relative area under disease progress curves (rAUDPC) of 10 anthracnose resistant and 10 susceptible sorghum genotypes, including Btx623 among 366 genotypes evaluated in three seasons in Ethiopia.**

decreased from K = 2 to K = 8, and increased steadily to a higher level at K = 20 (Fig 3). The optimal number of sub-populations were at K = 8 (Fig 4).

Individual ancestry was estimated for different K-values. Each cluster consisted of a genetically diverse and variable number of entries. About 65% of the assessed genotypes (203

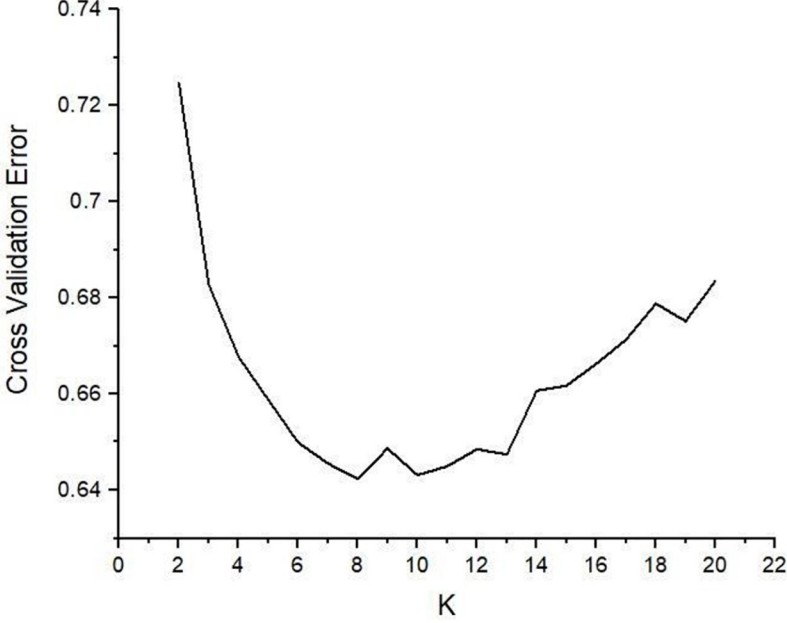

**Fig 3. Plot depicting the cross-validation error rates values and K sub-sets varying from K = 2 to K = 20 based on ADMIXTURE analysis.**

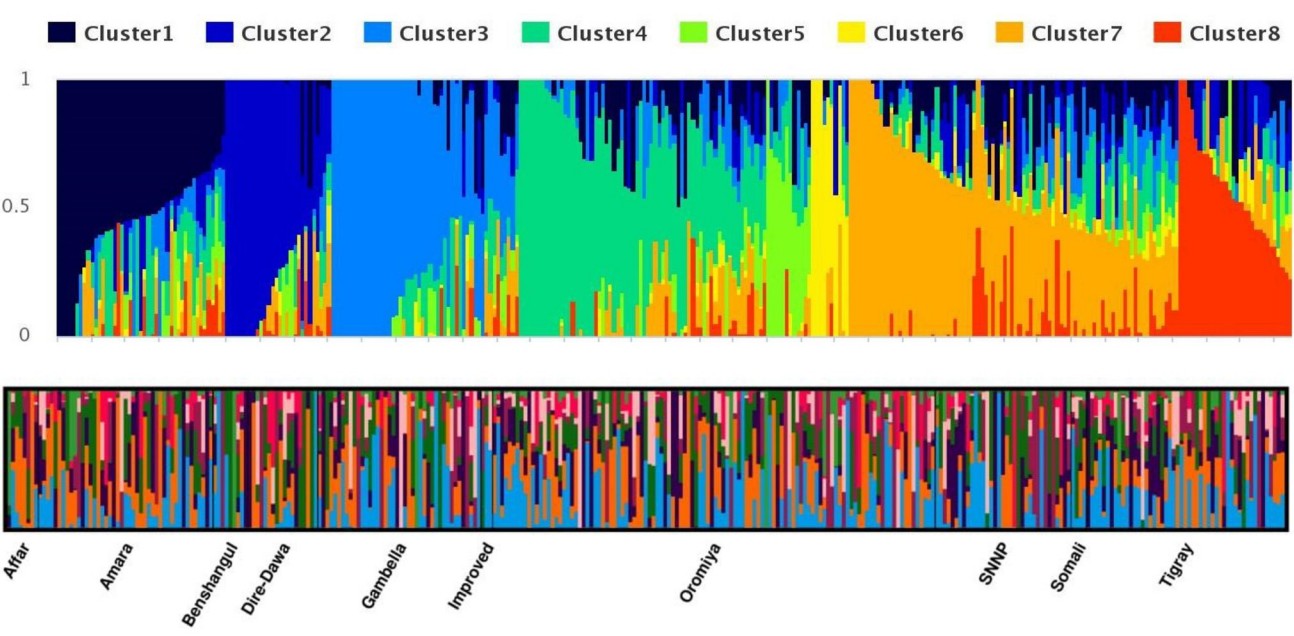

**Fig 4. Population genetic structure among 313 Ethiopian sorghum accessions in K = 8.**

accessions) were allocated in one population with a high ancestry membership coefficient with a likelihood of more than 0.60. The remaining 35% of test populations represented accessions with high admixture, which is expected from landrace populations.

Based on the probability of each genotype being allocated to one of the eight different groups, 43 genotypes (13%) fell into the seventh clusters (C-VII), 41 (12%) assigned to the third and four clusters (C-III and C-IV), and 32 (10%) into the first cluster (C-I), while 24 (7%), 17(6%), 10(3%), and 8(2%) genotypes were assigned to clusters 2, 8, 6, and 5, respectively. The remaining 114 genotypes (35%) were assigned to the admixed groups (Table 3).

A scree plot was generated to visualize the number of principal components. Overall, 10 principal components were identified of which principal components 1 (PC1), PC2 and PC3 explained relatively the highest proportion to the total variance (Fig 5A). Principal component analysis (PCA) using PC1 and PC2 stratified the test populations based on areas of collection (Fig 5B). The allocation of test genotypes were irrespective to the origin of collections.

## Marker-trait association

A summary of the genome-wide association analysis of anthracnose resistance and SNP markers amongst 313 sorghum collections is presented in Table 4. Eight significant ($P < 0.001$) marker-trait associations were resolved on chromosomes 1, 4, 6, 8, 9 and 10 (Table 4 and Fig 6A). The quantile-quantile plots (Fig 6B) of p-values were examined to determine how well the models accounted for population structure and familial relatedness.

Three markers, including *rs1887698* and *rs2681689* located on chromosome 9 and *rs1884746* on chromosome 10 had a significant ($P < 0.001$) association with anthracnose resistance. Markers *rs1887698* and *rs2681689* are significantly associated with chromosome 9 with $P = 4.4 \times 10^{-06}$ and $4.99 \times 10^{-06}$, respectively, explaining 15.9% of the total variation for anthracnose resistance. Whereas marker *rs1884746* located on chromosome 10 ($P = 4.28 \times 10^{-05}$) explained 15.0% of the total variation. Furthermore, SNPs significantly associated with anthracnose resistance were detected on chromosomes 1, 8, 9, 6 and 4, which explained 4.86 to

**Table 3. The proportion of the membership of each predefined population in each of the clusters obtained at optimum K (K = 8).**

| Population | Number of accessions | Admixed individuals (%) | The proportion of membership in each cluster (%) | | | | | | | |
|---|---|---|---|---|---|---|---|---|---|---|
| | | | C-I | C-II | C-III | C-IV | C-V | C-VI | C-VII | C-VIII |
| Afar | 8 | 25 | 0 | 13 | 13 | 36 | 0 | 13 | 0 | 0 |
| Amhara | 44 | 31 | 11 | 5 | 13 | 20 | 4 | 4 | 9 | 3 |
| Benishangul | 2 | 11 | 22 | 11 | 11 | 12 | 0 | 11 | 22 | 0 |
| Dire-Dawa | 18 | 22 | 22 | 11 | 11 | 16 | 6 | 0 | 6 | 6 |
| Gambella | 37 | 31 | 10 | 10 | 15 | 11 | 0 | 2 | 15 | 6 |
| Improved | 3 | 0 | 0 | 33 | 0 | 0 | 0 | 0 | 67 | 0 |
| Oromia | 108 | 39 | 4 | 6 | 8 | 14 | 2 | 4 | 14 | 9 |
| SNNP | 34 | 29 | 17 | 11 | 20 | 3 | 6 | 0 | 11 | 3 |
| Somali | 6 | 50 | 50 | 0 | 0 | 0 | 0 | 0 | 0 | 0 |
| Tigray | 53 | 47 | 6 | 4 | 15 | 8 | 2 | 2 | 14 | 2 |
| Total | 313 | 35 | 10 | 7 | 12 | 12 | 2 | 3 | 13 | 6 |

C-I = Cluster I, C-II = Cluster II, C-III = Cluster III, C-IV = Cluster IV, C-V = Cluster V, C-VI = Cluster VI, C-VII = Cluster VII, C-VII = Cluster VIII.

5.23% of the total variation among the tested sorghum collections for anthracnose resistance. Overall, eight SNPs were found to affect anthracnose resistance (Table 4), of which four SNPs were above the cutoff point (Fig 6A). Hence, the significant MTAs localized on the specific chromosomes influence anthracnose resistance and could be exploited in gene introgression and selection. Also, the proportion of the phenotypic variation ($R^2 > 4$) observed for all significant markers suggests their possible influence on anthracnose resistance.

## Putative genes that condition anthracnose resistance

A blast search of potential anthracnose resistance genes identified three candidate genes. The two candidate genes were located on chromosome 9, while one gene on chromosome 10. The two genes on chromosome 9 are annotated with protein-coding genes, including *Sobic.009G008800* (*xylem cysteine proteases*) and *Sobic.009G126300* (*Threonine-specific protein*

(A)   (B)

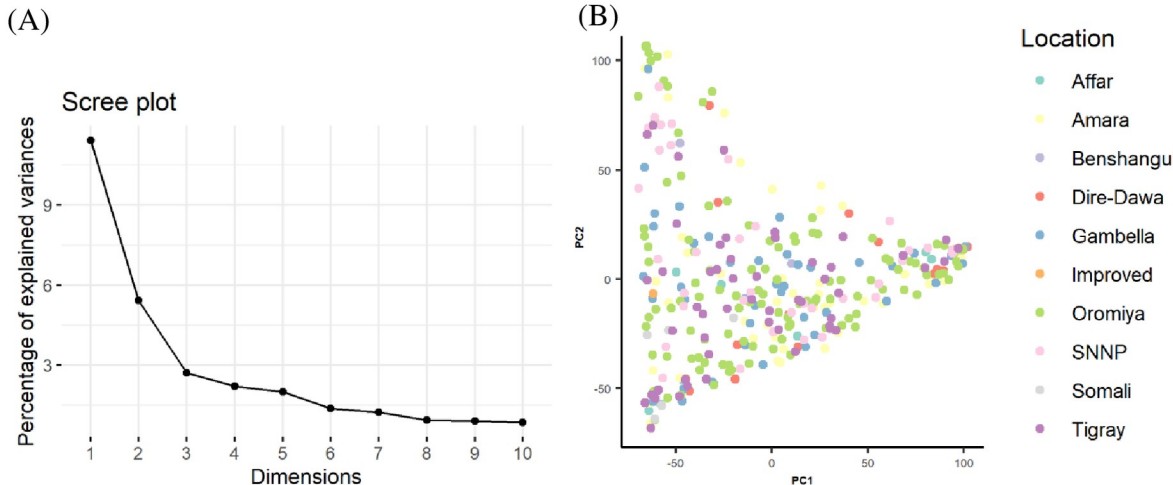

**Fig 5. Principal component analysis among 313 sorghum collections based on 11,643 SNPs using the first two principal components.** The large proportions of the variances contained in the data are retained by the first three principal components (A), while the relationship among collections between areas of origin is represented in (B).

**Table 4.  Summary on genome-wide association analysis of anthracnose resistance and SNP markers amongst 313 sorghum collections indicating significant markers, alleles detected with chromosome number and position, significant value, coefficient of determination ($R^2$) and annotation.**

| SNP ID | Allele | Chr | Position | P- value | $R^2$ | Gene position | | Gene names | Description of genes |
|---|---|---|---|---|---|---|---|---|---|
| | | | | | | Start | End | | |
| rs1887698 | A/G | 9 | 48008631 | 4.40E-06 | 15.9 | 48006606 | 48021494 | Sobic.009G126300 | Threonine-specific protein kinase |
| rs2681689 | C/T | 9 | 815114 | 4.99E-06 | 15.9 | 814415 | 816070 | Sobic.009G008800 | xylem cysteine proteases |
| rs100028710 | A/C | 4 | 5.30E+07 | 1.59E-05 | 5.23 | - | - | - | - |
| rs1938969 | C/T | 6 | 1561991 | 0.00068 | 5.21 | - | - | - | - |
| rs1884746 | A/G | 10 | 1055302 | 4.28E-05 | 15 | 1054558 | 1056549 | Sobic.010G012200 | Gluconokinase/Gluconate kinase |
| rs5196058 | C/T | 9 | 4.80E+07 | 0.00074 | 4.95 | - | - | - | - |
| rs2205151 | T/G | 8 | 5.40E+07 | 0.00032 | 4.87 | - | - | - | - |
| rs100052771 | A/G | 1 | 2.20E+07 | 0.00032 | 4.86 | - | - | - | - |

• denote not available.

*kinase)* linked to markers rs2681689 and rs1887698, respectively. In addition, one gene was annotated with protein-coding gene *Sobic.010G012200 (Gluconokinase/Gluconate kinase)* and linked to marker rs1884746 located on chromosome 10. The SNP data set used in the study is presented in S2 Table.

## Discussion

The existence of significant genotype by season interaction effects on anthracnose resistance suggests considerable genetic diversity among the assessed genotypes, and genotype performance varies across seasons. These results support earlier studies that there is potentially adequate genetic variation among Ethiopian sorghum to select for anthracnose resistance and to support genetic analysis, gene discovery and breeding efforts [8,9,11,38].

Genetic management of sorghum anthracnose requires unique sources of resistance for breeding programs. To identify new sources of anthracnose resistance genes, this study determined the population structure and association of genomic regions with anthracnose resistance. The studied sorghum population represents a diverse population of sorghum collections from Ethiopia, it's center of origin and diversity. Sorghum anthracnose resistance present in the test population was discerned through rigorous field screening in three seasons. Previous studies assessed the genetic basis of anthracnose resistance in sorghum [7]. The authors reported three loci on chromosome 5 amongst 335 US accessions using GWAS. In addition, Prom et al. (2019) [10] used 359 sorghum populations and identified quantitative trait loci on chromosome 8 conditioning resistance to anthracnose.

The population structure generated using hierarchical clustering, admixture, and principal component analysis identified a clear differentiation of the assessed sorghum collections (Figs 4 and 5). The test population was grouped into eight distinct genetic groups (Fig 4).

Previous studies of population structure identified 11 genetic groups using 1,425 Ethiopian sorghum accessions [12], while 374 accessions from Ethiopia were separated into 11 populations [39]. In addition, a total of 318 Sudanese sorghum core collections were evaluated with 183,144 SNP markers using the model-based clustering method that portrayed five subpopulations [11]. Furthermore, 940 diverse sorghum landraces of Ethiopia were assessed using 54,080 SNP markers that identified 12 subpopulations [40].

Markers associated with anthracnose resistance were identified on chromosomes 9 and 10, suggesting the value of these genomic regions in gene introgression and pyramiding programs. Previously, markers for anthracnose resistance in sorghum collections were reported in the US

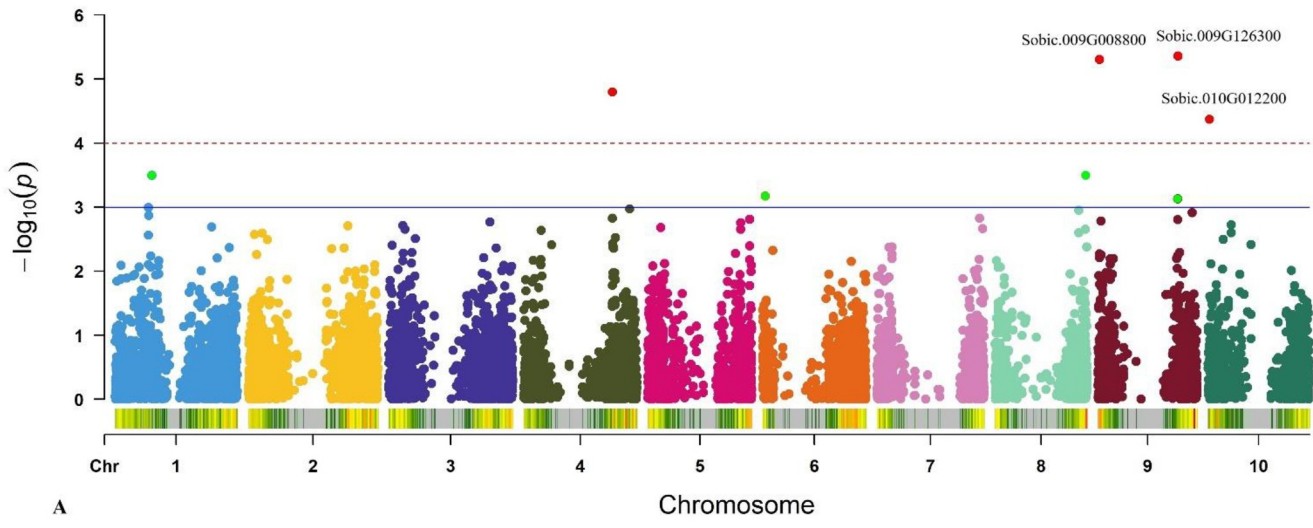

**Fig 6. Genome-wide association of anthracnose resistance amongst 313 sorghum collections with 11,643 SNP markers using a FarmCPU Model: (a) Manhattan plots showing significant false discovery rate (FDR) adjusted P-value of ≤0.1 associated with anthracnose resistance, A dash line represents the threshold from the FDR, and a blue line represents the significant threshold −Log10 (P) value and (b) Log Q-Q plots validating the FarmCPU Model and depicting consistency in reducing -log10(p-values) toward the expected level.**

sorghum collections. These markers were identified on chromosome 5 [7,9] and chromosome 9 [8] using SNP markers. The new markers are potentially useful for marker-assisted breeding of anthracnose resistance in the assessed sorghum populations [41].

The current study identified eight significant (P < 0.001) marker-trait associations on chromosomes 1, 4, 6, 8, 9 and 10 (Table 4 and Fig 6A). The quantile-quantile plots (Fig 6B) showed a perfect fit of the observed and expected population structures for anthracnose resistance using the FarmCPU Model. The novel genes are additions to previously identified genomic regions conditioning resistance to anthracnose, downy mildew, and head smut in sorghum [4,7–9]. Four SNP markers above the cutoff point (Fig 6A) are novel anthracnose resistance markers detected in this study. The markers are located on chromosomes 9, 10 and 4. Previously [4] used 242 mini core accessions and three sorghum cultivars and were subjected to genome-wide association analysis. The authors reported the anthracnose resistance gene on chromosome 8. In addition [9], examined 114 recombinant inbred lines (RILs) obtained from crosses of sorghum anthracnose resistant line SC112-14 (PI533918) and susceptible line PI609251 and identified tightly linked anthracnose resistance locus on chromosome 5.

Through blast analysis, the present study identified three candidate anthracnose resistance genes. The two genes are located on chromosome 9, while one gene is on chromosome 10. The two genes located on chromosome 9 are annotated with protein-coding genes *Sobic.009G008800* and *Sobic.009G126300* linked to markers rs1887698 and rs2681689, respectively. The *Sobic.009G008800* reportedly had a strong association with annotated function xylem cysteine peptidase 1 of sorghum conditioning disease resistance. The biological effect of the gene includes programmed cell death, responsive to dehydration, while the annotated sorghum genes had roles in anthracnose resistance on chromosome 9 [42,43]. Pogány et al. (2015) [42] reported that cysteine proteases is one of the first Arabidopsis proteases functioning in the immune system of sorghum. In a mutant strain of Arabidopsis, overexpression of the aspartic protease Constitutive Disease Resistance 1 (CDR1) resulted in resistance to the virulent strain of *Pseudomonas syringae*, a bacterium that causes diseases of monocots, herbaceous dicots, and woody dicots. Cysteine proteases also contain a cysteine nucleophilic residue that performs a nucleophilic attack proteolysis resulting in an intermediate state where the enzyme is covalently attached to its substrate [44]. Cysteine proteases reportedly conferred immunity in tomatoes against diseases caused by *Phytophthora infestans* (Pinf) and *Cladosporium fulvum* (syn. Passalora fulva) [45,46].

In the present study, a second significant candidate gene, *Sobic.009G126300* was located on chromosome 9 linked to marker rs1887698 with annotated function similar to Serine/threonine-protein kinase CTR1. Serine/threonine-protein kinase is reportedly known for mediating drought stress tolerance in plants. Furthermore, the threonine-protein kinase is believed to have a role in regulating ethylene hormone pathways conditioning drought response in senna (*Cassia angustifolia* Vahl.) [47]. The 3rd candidate gene identified in this study is linked to the marker rs1884746 annotated with *Sobic.010G012200 (Gluconokinase/Gluconate kinase)*. Gluconate kinase is an essential enzyme of the oxidative pentose phosphate pathway responsible for NADPH supplies during fatty acid synthesis in developing embryos of *Thermotoga maritima* [48,49].

The anthracnose resistance candidate genes identified in the present study will be validated across multiple seasons and locations as ideal molecular markers for anthracnose resistance breeding programs. The novel genes could be stacked into anthracnose susceptible sorghum lines through marker-assisted or recurrent selection method. A combination of novel QTLs would render durable resistance to sorghum anthracnose.

## Conclusion

Marker trait association is key to identifying genomic regions associated with anthracnose resistance for marker-assisted breeding in sorghum improvement programs. The present study identified four novel markers associated with anthracnose resistance designated as rs1887698, rs2681689, rs1884746 and rs100028710. The new genetic markers identified in the current sorghum populations are valuable genomic resources for future parental selection, quantitative trait loci analysis, trait introgression, gene pyramiding and marker-assisted selection of anthracnose resistance in sorghum breeding programs in Ethiopia or related agro-ecologies. Further studies are required to validate the significant markers identified in the present study.

## Supporting information

**S1 Table. Mean FAS and rAUDPC values.**
(CSV)

**S2 Table. SNP data set.**
(CSV)

## Acknowledgments

The authors thank the Integrated Genotyping Service and Support (IGSS) platform of the Biosciences Eastern and Central Africa (BecA-ILRI)/Kenya is acknowledged for genotyping the sorghum accessions.

## Author Contributions

**Conceptualization:** Girma Mengistu, Hussein Shimelis, Dagnachew Lule.

**Data curation:** Girma Mengistu, Hussein Shimelis, Dagnachew Lule.

**Formal analysis:** Girma Mengistu, Hussein Shimelis, Ermias Assefa, Dagnachew Lule.

**Funding acquisition:** Girma Mengistu, Hussein Shimelis, Dagnachew Lule.

**Investigation:** Girma Mengistu, Hussein Shimelis, Dagnachew Lule.

**Methodology:** Girma Mengistu, Hussein Shimelis, Ermias Assefa, Dagnachew Lule.

**Project administration:** Girma Mengistu, Hussein Shimelis, Dagnachew Lule.

**Resources:** Girma Mengistu, Hussein Shimelis, Ermias Assefa, Dagnachew Lule.

**Software:** Girma Mengistu, Hussein Shimelis, Dagnachew Lule.

**Supervision:** Girma Mengistu, Hussein Shimelis, Dagnachew Lule.

**Validation:** Girma Mengistu, Hussein Shimelis, Dagnachew Lule.

**Visualization:** Girma Mengistu, Hussein Shimelis, Dagnachew Lule.

**Writing – original draft:** Girma Mengistu, Hussein Shimelis, Ermias Assefa.

**Writing – review & editing:** Girma Mengistu, Hussein Shimelis, Dagnachew Lule.

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
