## [Decision Letter · Decision Letter 0]

7 May 2021

PONE-D-21-10117

Genome-wide association analysis of anthracnose resistance in sorghum [Sorghum bicolor (L.) Moench]

PLOS ONE

Dear Dr. Digafe,

Thank you for submitting your manuscript to PLOS ONE. After careful consideration, we feel that it has merit but does not fully meet PLOS ONE’s publication criteria as it currently stands. Therefore, we invite you to submit a revised version of the manuscript that addresses the points raised during the review process.

We look forward to receiving your revised manuscript.

Kind regards,

Karthikeyan Adhimoolam

Academic Editor

PLOS ONE

Journal Requirements:

Reviewers' comments:

Reviewer's Responses to Questions

**Comments to the Author**

1. Is the manuscript technically sound, and do the data support the conclusions?

Reviewer #1: Partly

Reviewer #2: Yes

2. Has the statistical analysis been performed appropriately and rigorously? 

Reviewer #1: Yes

Reviewer #2: Yes

3. Have the authors made all data underlying the findings in their manuscript fully available?

Reviewer #1: No

Reviewer #2: Yes

4. Is the manuscript presented in an intelligible fashion and written in standard English?

Reviewer #1: No

Reviewer #2: Yes

5. Review Comments to the Author

Reviewer #1: Genome wide association analysis of anthracnose resistance in sorghum [Sorghum bicolor (L.) Moench]

This manuscript includes the anthracnose resistance evaluation of 366 Ethiopian accessions over three years and the genotyping characterization of these accessions based on DArT sequencing markers. Later the authors analyzed the population structure of this Ethiopian accessions and performed GWAS to identify anthracnose resistance loci.

Definitely, this manuscript has valuable information for sorghum breeding program in Ethiopia and other countries. However, the authors need to improve the manuscript to be accepted for publications. At this stage, the manuscript has many concerns that need to be addressed, and I suggest to the authors to resubmit the manuscript after its improvement.

1- The English of the manuscript need to be improved. It has multiple typo errors and sentences that need to be rewrite for clarification. The use of number in reference needs to be verified in the sentences because many are difficult to read. Eg. Also [9] examined 114 recombinant inbred lines……

2- The evaluation of anthracnose is not well explained in the manuscript. They just mention is according to Chala et al. 2012. They need to provide some information of how they did the scoring.

3- Candidate genes were identified based in the SNPs position. They authors should explain the window size (20Kb upstream/downstream of the SNP) used to identity candidate genes.

4- They authors should provide a supplementary table with the anthracnose score of each accession. Also make available the DArT genotyping.

5- The authors never mentioned how many accessions were finally classified as resistant to anthracnose. I am wondering if they did a means compare analysis, contrast, or any other statistical analysis to classify an accession resistant to anthracnose.

6- Population structure analysis is confuse. They first mention the optimal number is 8, then in the second paragraph they mention 10.

7- The marker trait association is confuse. In Mat and Met they indicate they will use the FDR p values less than 0.1 according to Benjamini and Hochberg. Then in results they mentioned eight significant SNPs and highlight that 4 were above the cutoff point. Do you have two threshold? You mentioned just one in Mat and Met.

8- The discussion is poor. Previous studies based on Ethiopian germplasm identified an anthracnose resistant locus in chromosome 9. How distance is the DArT markers from this locus?

9- The discussion don’t include the population structure analysis of other Ethiopian germplasm. Eg. Cuevas et al. 2017 BMC Genomics, Girma et al. 2019, Menamo et al. 2021 TAG,

10 – Most of the disscussion is centralized in candidate genes, instead of how to use the information for breeding and the conservation of sorghum germplasm

11- Reference num 38 have an incorrect format. Last name must be first.

Reviewer #2: The manuscript entitled “Genome-wide association analysis of anthracnose resistance in sorghum [Sorghum bicolor (L.) Moench]” was written well and identified four novel markers (rs1887698, rs2681689, rs1884746 and rs100028710) associated with anthracnose resistance. Also by using the reference genome, authors identified the putative functional candidate genes annotated with genes in other species such as Arabidopsis and tomato. Population structure analysis with the clustering of lines and GWAS study for marker-trait association is well documented. However, the manuscript will be completed only if the identified markers are validated with additional study. To accelerate the breeding process for anthracnose resistance, only validated markers are more useful for large scale screening. Hence, authors are requested to provide the additional information on the identified markers’ validation.

6. PLOS authors have the option to publish the peer review history of their article (what does this mean?). If published, this will include your full peer review and any attached files.

Reviewer #1: No

Reviewer #2: No

---

## [Author Response · Author response to Decision Letter 0]

27 Jun 2021

Manuscript number, PONE-D-21-10117

Reviewers' comments:

Reviewer's Responses to Questions

Comments to the Author

1. Is the manuscript technically sound, and do the data support the conclusions?

Reviewer #1: Partly

Reviewer #2: Yes

Response 1. Thank you for the constructive feedback. The manuscript is based on integrated data sets through field-based phenomic and genomic analyses which supported the final conclusions and recommendations. The field experiments were conducted through rigorous data location across three consecutive seasons involving sufficiently large number of sorghum germplasm collections and anthracnose susceptible and resistant standard controls. The conclusions drawn from the study were in synchronization with the phenomic and genomic data and sound genetic analyses. 

2. Has the statistical analysis been performed appropriately and rigorously?

Reviewer #1: Yes

Reviewer #2: Yes

Response 2. Thank you for the comment. The statistical analysis has been performed appropriately and rigorously.

3. Have the authors made all data underlying the findings in their manuscript fully available?

Reviewer #1: No

Reviewer #2: Yes

Response 3. The comment is accepted. Both data sets (genotypic and anthracnose severity data) are provided as a Supplementary Table 1 and 2. ________________________________________

 4. Is the manuscript presented in an intelligible fashion and written in standard English?

 Reviewer #1: No

Reviewer #2: Yes

Response 4. Thank you for the comments. The authors have checked the manuscript and typographical or grammatical errors are promptly fixed.________________________________________

5. Review Comments to the Author

Reviewer #1: Genome wide association analysis of anthracnose resistance in sorghum [Sorghum bicolor (L.) Moench]

This manuscript includes the anthracnose resistance evaluation of 366 Ethiopian accessions over three years and the genotyping characterization of these accessions based on DArT sequencing markers. Later the authors analyzed the population structure of this Ethiopian accessions and performed GWAS to identify anthracnose resistance loci.

Definitely, this manuscript has valuable information for sorghum breeding program in Ethiopia and other countries. However, the authors need to improve the manuscript to be accepted for publications. At this stage, the manuscript has many concerns that need to be addressed, and I suggest to the authors to resubmit the manuscript after its improvement.

1- The English of the manuscript need to be improved. It has multiple typo errors and sentences that need to be rewrite for clarification. The use of number in reference needs to be verified in the sentences because many are difficult to read. Eg. Also [9] examined 114 recombinant inbred lines……

Response 1. As advised, the use of number(s) in the reference have been revised. This will ease to reading the references in the manuscript.

2- The evaluation of anthracnose is not well explained in the manuscript. They just mention is acording to Chala et al. 2012. They need to provide some information of how they did the scoring.

Response 2. The comment is accepted. We have added one paragraph.

3- Candidate genes were identified based in the SNPs position. They authors should explain the window size (20Kb upstream/downstream of the SNP) used to identity candidate genes.

Response 3. Thank you for the insight. This was achieved using the physical genome assembly of sorghum reference genome sequence, version 3.1.1 (https://phytozome.jgi.doe.gov/pz/portal.html#!info?alias=Org Sbicolor) serving as identifying candidate genes between 20 kb on either side of significant SNPs.

4- They authors should provide a supplementary table with the anthracnose score of each accession. Also make available the DArT genotyping.

Response 4. Data on anthracnose score for each accession and DArT genotyping are provided as Supplementary Table 1 and 2.

5- The authors never mentioned how many accessions were finally classified as resistant to anthracnose. I am wondering if they did a means compare analysis, contrast, or any other statistical analysis to classify an accession resistant to anthracnose.

Response 5. Thank you for the comment. 63 accessions were identified as moderately resistant to anthracnose disease. This is indicated in the Results section.

6- Population structure analysis is confuse. They first mention the optimal number is 8, then in the second paragraph they mention 10.

Response 6. The error is corrected, and the number of sub-populations are 8.

7- The marker trait association is confuse. In Mat and Met they indicate they will use the FDR p values less than 0.1 according to Benjamini and Hochberg. Then in results they mentioned eight significant SNPs and highlight that 4 were above the cutoff point. Do you have two threshold? You mentioned just one in Mat and Met.

Response 7. Thank you. We have added a description in Fig 4. A dash line represents the threshold from the FDR, and a blue line represents the significant threshold −Log10 (P) value.

8- The discussion is poor. Previous studies based on Ethiopian germplasm identified an anthracnose resistant locus in chromosome 9. How distance is the DArT markers from this locus?

Response 8. The distance from previously identified Ethiopian germplasm an anthracnose resistant locus in chromosome 9 is 360Kb.

9- The discussion don’t include the population structure analysis of other Ethiopian germplasm. Eg. Cuevas et al. 2017 BMC Genomics, Girma et al. 2019, Menamo et al. 2021 TAG,

Response 9. Thank you for this input. We have revised as follows: Previous studies of population structure identified 11 genetic groups using 1,425 Ethiopian sorghum accessions [12]. In addition, a total of 318 Sudanese sorghum core collections were evaluated with 183,144 SNP markers using the model-based clustering method that portrayed five subpopulations [11]. Furthermore, 940 diverse sorghum landraces of Ethiopia were assessed using 54,080 SNP markers that identified 12 subpopulations [48].

10 – Most of the discussion is centralized in candidate genes, instead of how to use the information for breeding and the conservation of sorghum germplasm.

Response 10. The anthracnose resistance genes identified in the present study will be validated across multiple seasons and locations to serves as ideal molecular markers for anthracnose resistance breeding programs. The novel genes could be pyramided into anthracnose susceptible sorghum lines through marker-assisted selection. A combinations of QTLs could render durable resistance to sorghum anthracnose.

11- Reference num 38 have an incorrect format. Last name must be first.

Response 11. Accepted. We have corrected the reference number 38 in the manuscript.

Reviewer #2: The manuscript entitled “Genome-wide association analysis of anthracnose resistance in sorghum [Sorghum bicolor (L.) Moench]” was written well and identified four novel markers (rs1887698, rs2681689, rs1884746 and rs100028710) associated with anthracnose resistance. Also by using the reference genome, authors identified the putative functional candidate genes annotated with genes in other species such as Arabidopsis and tomato. Population structure analysis with the clustering of lines and GWAS study for marker-trait association is well documented. However, the manuscript will be completed only if the identified markers are validated with additional study. To accelerate the breeding process for anthracnose resistance, only validated markers are more useful for large scale screening. Hence, authors are requested to provide the additional information on the identified markers’ validation.

Response. Thank you for the encouraging comment. Our next plan is to validate the identified four novel markers and implement marker-assisted breeding to stack the genes in a desirable genetic background. 

6. PLOS authors have the option to publish the peer review history of their article (what does this mean?). If published, this will include your full peer review and any attached files.

Response 6. Yes

Do you want your identity to be public for this peer review? For information about this choice, including consent withdrawal, please see our Privacy Policy.

Reviewer #1: No

Reviewer #2: No

---

## [Decision Letter · Decision Letter 1]

14 Sep 2021

PONE-D-21-10117R1Genome-wide association analysis of anthracnose resistance in sorghum [Sorghum bicolor (L.) Moench]PLOS ONE

Dear Dr. Girma

Thank you for submitting your manuscript to PLOS ONE. After careful consideration, we feel that it has merit but does not fully meet PLOS ONE’s publication criteria as it currently stands. Therefore, we invite you to submit a revised version of the manuscript that addresses the points raised during the review process.

We look forward to receiving your revised manuscript.

Kind regards,

Karthikeyan Adhimoolam

Academic Editor

PLOS ONE

Journal Requirements:

Additional Editor Comments (if provided):

Recommended for minor revision.

Reviewers' comments:

Reviewer's Responses to Questions

**Comments to the Author**

1. If the authors have adequately addressed your comments raised in a previous round of review and you feel that this manuscript is now acceptable for publication, you may indicate that here to bypass the “Comments to the Author” section, enter your conflict of interest statement in the “Confidential to Editor” section, and submit your "Accept" recommendation.

Reviewer #1: (No Response)

Reviewer #3: (No Response)

2. Is the manuscript technically sound, and do the data support the conclusions?

Reviewer #1: Yes

Reviewer #3: Yes

3. Has the statistical analysis been performed appropriately and rigorously? 

Reviewer #1: Yes

Reviewer #3: Yes

4. Have the authors made all data underlying the findings in their manuscript fully available?

Reviewer #1: Yes

Reviewer #3: No

5. Is the manuscript presented in an intelligible fashion and written in standard English?

Reviewer #1: Yes

Reviewer #3: Yes

6. Review Comments to the Author

Reviewer #1: Dear author,

The manuscript improved from the first revision, however, it still have some concerns that need to be addressed.

1- The revised version still having error with the reference and gramatical. Some edits that were red in revised version are not integrated in the clean version. Please verify the whole document.

Eg. In addition, [9] examined…………but in the edit section look that should be fixed by In addition, Cruet-Burgos et al. [9].

2- FarmCPU don’t use kinship. Please delete the sentence in Mat and Met.

3- The 20 Kb window is arbitrary and large to identified candidate genes. If you don’t have information about how large is the LD block region (I was expecting that as an answer), you need to be carefully in how to select candidate genes. I suggest you should mention: 1) the physical distance among the associate SNP and candidate gene, 2) Are any other SNP close to these candidate genes?

4- The figure 4 is meaningless. I understand your objective but the mixed among Population structure and collection are make the figure impossible to understand. I suggest take out collection area present the disrupting of population structure from K2 to K8.

You might consider construct other type of graph to determine the relationship among population structure and collection site. Based on what you present is not association, and that is also observed in the PC graph.

5- The blue line in the Fig 6 represent the significant threshold for p <0.001…no the threshold for -Log10 (P) value..You can mention -Log10 p (0.001). Same in Mat and Met’

6- In the discussion, you mention about previous population structure analysis. You include the NPGS Sudan collection study [11] but not mention the population structure of NPGS Ethiopian collection (Cuevas et al. 2017. BMC Genomic) which is more related to your work.

7- I think the discussion still need to be improved. At least I suggest you could explain: 1) why the population structure you found (8 populations) do not have association with the collection site of the samples.

Reviewer #3: The Manuscript entitled "Genome-wide association analysis of anthracnose resistance in sorghum [Sorghum bicolor (L.) Moench]" well written except discussion part. In the discussion, Please provide the information on how the identified 4 markers can be utilised for the resistance breeding program, which is key to the readers. Moreover, the author need to do correction, whereever marked. Furthermore, please provide the information on disease scoing part in details and attach the scoring details for each entry as supplementary table. However, the minor revision is needed for the publication acceptance.

7. PLOS authors have the option to publish the peer review history of their article (what does this mean?). If published, this will include your full peer review and any attached files.

Reviewer #1: No

Reviewer #3: No

---

## [Author Response · Author response to Decision Letter 1]

28 Oct 2021

Manuscript number, PONE-D-21-10117

Reviewers' comments:

Reviewer's Responses to Questions

Comments to the Author

1. If the authors have adequately addressed your comments raised in a previous round of review and you feel that this manuscript is now acceptable for publication, you may indicate that here to bypass the “Comments to the Author” section, enter your conflict of interest statement in the “Confidential to Editor” section, and submit your "Accept" recommendation.

Reviewer #1: (No Response)

Reviewer #3: (No Response)

Response 1. Thank you for the constructive feedback. 

2. Is the manuscript technically sound, and do the data support the conclusions?

Reviewer #1: Yes

Reviewer #3: Yes

Response 2. Thank you. 

 3. Has the statistical analysis been performed appropriately and rigorously?

Reviewer #1: Yes

Reviewer #3: Yes

Response 3. Thank you. ________________________________________

4. Have the authors made all data underlying the findings in their manuscript fully available?

Reviewer #1: Yes

Reviewer #3: No

Response 4. As required, both data sets (genotypic and anthracnose severity data) are provided as supplementary Tables 1 and 2.

5. Is the manuscript presented in an intelligible fashion and written in standard English?

Reviewer #1: Yes

Reviewer #3: Yes 

Response 5. Thank you. 

6. Review Comments to the Author

Reviewer #1: Dear author, 

Reviewer #1: 

1- The revised version still having error with the reference and gramatical. Some edits that were red in revised version are not integrated in the clean version. Please verify the whole document.

Eg. In addition, [9] examined…………but in the edit section look that should be fixed by In addition, Cruet-Burgos et al. [9].

Response 1. Thank you for the feedback. As advised, the document is reviewed for grammatical and reference errors. 

This reference is corrected in text: In addition, [9] examined…………is replaced with In addition, Cruet-Burgos et al. [9].

2- FarmCPU don’t use kinship. Please delete the sentence in Mat and Met.

Response 2. Accepted. Deleted from the Materials and Methods

3- The 20 Kb window is arbitrary and large to identified candidate genes. If you don’t have information about how large is the LD block region (I was expecting that as an answer), you need to be carefully in how to select candidate genes. I suggest you should mention: 1) the physical distance among the associate SNP and candidate gene, 2) Are any other SNP close to these candidate genes?

Response 3. Thank you for the insight. 

At a physical distance of 51.4 kbp, the fitted LOESS curve intersected with the critical LD value. The LD values below this threshold were assumed to be due to physical linkage among inter-SNP pairs. However, in the present association panel, the 20 kb (above and below the significant SNPs) window, was chosen to be within the anticipated window of LD decay. The blue line is the trend line of nonlinear regressions against physical distance. The crucial value of r2 (0.1) and the LD decay value are represented by the horizontal red and vertical green lines, respectively.

4- The figure 4 is meaningless. I understand your objective but the mixed among Population structure and collection are make the figure impossible to understand. I suggest take out collection area present the disrupting of population structure from K2 to K8.

You might consider construct other type of graph to determine the relationship among population structure and collection site. Based on what you present is not association, and that is also observed in the PC graph.

Response 4. The population structure from K2 to K8 is removed, and a new figure has replaced this.

5- The blue line in the Fig 6 represent the significant threshold for p <0.001…no the threshold for -Log10 (P) value.You can mention -Log10 p (0.001). Same in Mat and Met’

Response 5. Thank you for the comment. A sentence is added in Material and Method section. An exploratory significance cutoff p <0.001 was also used and p <0.001 was also added in figure 6.

6- In the discussion, you mention about previous population structure analysis. You include the NPGS Sudan collection study [11] but not mention the population structure of NPGS Ethiopian collection (Cuevas et al. 2017. BMC Genomic) which is more related to your work.

Response 6. Accepted. The population structure of NPGS Ethiopian collection is mentioned in the discussion. 

7- I think the discussion still need to be improved. At least I suggest you could explain: 1) why the population structure you found (8 populations) do not have association with the collection site of the samples.

Response 7. The population structure was independent of the collection sites, given the higher chance of genetic admixtures due to open-pollination that vary from 5 to 15% in sorghum genotypes (ADD REFERENCE). 

Reviewer #3: The Manuscript entitled "Genome-wide association analysis of anthracnose resistance in sorghum [Sorghum bicolor (L.) Moench]" well written except discussion part. In the discussion, Please provide the information on how the identified 4 markers can be utilised for the resistance breeding program, which is key to the readers. Moreover, the author need to do correction, whereever marked. Furthermore, please provide the information on disease scoing part in details and attach the scoring details for each entry as supplementary table. However, the minor revision is needed for the publication acceptance.

Response. Thank you for the encouraging comment. This is concisely summarized in the discussion indicating the value and use of the 4 new markers as follows: 

“The anthracnose resistance candidate genes identified in the present study will be validated across multiple seasons and locations to serve as ideal molecular markers for anthracnose resistance breeding programs. The novel genes could be stacked into anthracnose susceptible sorghum lines through marker-assisted or recurrent selection method. A combination of novel QTLs would render durable resistance to sorghum anthracnose.” 

 6. PLOS authors have the option to publish the peer review history of their article (what does this mean?). If published, this will include your full peer review and any attached files.

Response 6. Yes

Do you want your identity to be public for this peer review? For information about this choice, including consent withdrawal, please see our Privacy Policy.

Reviewer #1: No

Reviewer #2: No

---

## [Decision Letter · Decision Letter 2]

3 Dec 2021

Genome-wide association analysis of anthracnose resistance in sorghum [Sorghum bicolor (L.) Moench]

PONE-D-21-10117R2

Dear Dr. Girma Mengistu Digafe,

We’re pleased to inform you that your manuscript has been judged scientifically suitable for publication and will be formally accepted for publication once it meets all outstanding technical requirements.

Kind regards,

Karthikeyan Adhimoolam

Academic Editor

PLOS ONE

Additional Editor Comments (optional):

-nil-

Reviewers' comments:

Reviewer's Responses to Questions

**Comments to the Author**

1. If the authors have adequately addressed your comments raised in a previous round of review and you feel that this manuscript is now acceptable for publication, you may indicate that here to bypass the “Comments to the Author” section, enter your conflict of interest statement in the “Confidential to Editor” section, and submit your "Accept" recommendation.

Reviewer #3: All comments have been addressed

2. Is the manuscript technically sound, and do the data support the conclusions?

Reviewer #3: Yes

3. Has the statistical analysis been performed appropriately and rigorously? 

Reviewer #3: Yes

4. Have the authors made all data underlying the findings in their manuscript fully available?

Reviewer #3: Yes

5. Is the manuscript presented in an intelligible fashion and written in standard English?

Reviewer #3: Yes

6. Review Comments to the Author

Reviewer #3: "Genome-wide association analysis of anthracnose resistance in sorghum [Sorghum bicolor (L.) Moench]" well written and gave excellent presentation on analytical part. Revised manuscript addressed the previous correction and adequately given the detailed notes on the discussion part. Hence, i am concuding that this manuscript can be accepted for the publication.

7. PLOS authors have the option to publish the peer review history of their article (what does this mean?). If published, this will include your full peer review and any attached files.

Reviewer #3: No

---

## [Editor Report · Acceptance letter]

9 Dec 2021

PONE-D-21-10117R2 

Genome-wide association analysis of anthracnose resistance in sorghum [*Sorghum bicolor* (L.) Moench] 

Dear Dr. Mengistu:

I'm pleased to inform you that your manuscript has been deemed suitable for publication in PLOS ONE. Congratulations! Your manuscript is now with our production department. 

Kind regards, 

on behalf of

Dr. Karthikeyan Adhimoolam 

Academic Editor

PLOS ONE